# The Utility of Faecal Calprotectin, Lactoferrin and Other Faecal Biomarkers in Discriminating Endoscopic Activity in Crohn’s Disease: A Systematic Review and Meta-Analysis

**DOI:** 10.3390/biomedicines11051408

**Published:** 2023-05-09

**Authors:** Anuj Bohra, Ghada Mohamed, Abhinav Vasudevan, Diana Lewis, Daniel R. Van Langenberg, Jonathan P. Segal

**Affiliations:** 1Department of Gastroenterology, Eastern Health, Box Hill, Melbourne, VIC 3128, Australia; 2Department of Gastroenterology, Northern Health, Epping, Melbourne, VIC 3076, Australia; 3Department of Gastroenterology, Duke University Health System, Durham, NC 27710, USA; 4Northern Health Clinical School, University of Melbourne, Epping, Melbourne, VIC 3076, Australia; 5Department of Gastroenterology, Royal Melbourne Hospital, Parkville, Melbourne, VIC 3050, Australia

**Keywords:** Crohns’ Disease, biomarkers, endoscopy, accuracy

## Abstract

Introduction: Currently, faecal calprotectin (FC) is the predominate faecal biomarker utilised in clinical practice to monitor Crohn’s disease (CD) activity. However, there are several potential faecal biomarkers described in the literature. We performed a meta-analysis to determine the accuracy of faecal biomarkers in discriminating endoscopic activity and mucosal healing in CD. Methods: We searched the medical literature using MEDLINE, EMBASE, and PubMed from 1978 to 8 August 2022. Descriptive statistics, including sensitivity, specificity of the primary studies, their positive and negative likelihood ratios, and their diagnostic odds ratio (DOR), were calculated. The methodological quality of the included studies was evaluated using the Quality Assessment of Diagnostic Accuracy Studies-2 (QUADAS) criteria. Results: The search found 2382 studies, of which 33 were included for analysis after screening. FC was found to have a pooled sensitivity and specificity, DOR, and negative predictive value (NPV) in discriminating active endoscopic disease (versus inactive) of 81%, 74%, 13.93, and 0.27, respectively. Faecal lactoferrin (FL) had a pooled sensitivity and specificity, DOR, and NPV in discriminating active endoscopic disease of 75%, 80%, 13.41, and 0.34, respectively. FC demonstrated a pooled sensitivity and specificity, DOR, and NPV of 88%, 72%, 18.17, and 0.19 in predicting mucosal healing. Conclusion: FC remains an accurate faecal biomarker. Further evaluation of the utility of novel faecal biomarkers is needed.

## 1. Introduction

Crohn’s disease (CD) is a chronic immune-inflammatory condition with a growing incidence in developed countries [1,2,3]. Whilst inflammatory changes including ulceration can manifest throughout the GIT, the most commonly affected luminal regions include the small bowel in 70% of patients and CD limited to the colon in 20% [4]. Left untreated, the inflammatory burden of CD can result in complications including stenosis formation and subsequent intestinal obstruction, fistula formation, as well as increased risk of infective complications such as intra-abdominal abscess and development of colorectal cancer [5,6]. Thus, current therapy for CD is geared toward an overall reduction in inflammatory burden and prevention of complications associated with longstanding disease [7,8,9,10]. Current treatment targets in CD include endoscopic remission assessed via the simplified endoscopic score for CD (SES-CD) and/or the CD endoscopic index of severity (CDEIS) and/or radiological targets such as transmural healing which is best evaluated via magnetic resonance enterography (MRE) and/or intestinal ultrasound (IUS) [11,12]).

Current STRIDE-2 guidelines recommend serial, objective assessment of disease activity at 6–12 monthly intervals [13,14]. Whilst ileocolonoscopic assessment may be desirable, multiple limitations preclude regular and repeated endoscopic assessment, including cost and resource constraints, the need for bowel preparation and sedation anaesthesia, plus procedural risks including bowel perforation and bleeding. Thus, there is a need for cheap, rapid, objective, and patient-friendly alternatives for disease assessment. Faecal biomarkers are attractive in this setting, and their capacity for discriminating endoscopic activity in CD has been studied extensively, thus promoting their utility in serial monitoring of CD activity. Currently, faecal calprotectin (FC) is the most commonly used faecal biomarker in CD. Calprotectin is a calcium- and zinc-binding protein of the s-100 protein family released predominantly by neutrophils that migrate to the small and large intestines during periods of active bowel inflammation [15,16]. The concentration of FC has been shown to correlate with active neutrophilic inflammation within the bowel, although it is not specific to CD or inflammatory bowel disease (IBD) [15,16]. Furthermore, a number of other faecal biomarkers of promising utility for discriminating endoscopic CD activity have been described in the literature with the potential to be incorporated more widely into routine clinical practice, including faecal lactoferrin (FL), faecal immunochemical test (FIT), neopterin, metalloprotease-9 (MMP-9), myeloperoxidase, faecal lipocalin-2 (FCLN-2), chitinase 3-like 1 (CHI3L1), polymorphonuclear neutrophil elastase (PMN-e), microRNA, S100A12, and alpha-1 antitrypsin. Therefore, this meta-analysis was conducted to assess and reaffirm the accuracy of FC and examine other studied faecal biomarkers in discriminating endoscopic activity in CD.

## 2. Materials and Methods

### 2.1. Literature Search

We searched the medical literature using MEDLINE (via Ovid), EMBASE, and PubMed from 1 January 1978 to 8 August 2022 inclusive. To identify published abstracts, we hand searched conference proceedings from the United European Gastroenterology Week, European Crohn’s and Colitis Organisation, British Society of Gastroenterology, and Digestive Diseases Week from September 2016 to August 2022 inclusive.

We searched the medical literature using the terms in the appendix using both medical subject headings [MeSH] and free-text terms. Only English language manuscripts were reviewed. We hand searched references from eligible studies and reviews for any further studies to be included. This process adhered to a standard, prespecified study protocol which was registered on the International Prospective Register of Systematic Reviews(PROSPERO) and given the study identification number: CRD42022354526 [17].

### 2.2. Study Selection

The search was uploaded to Covidence, a webtool for systematic reviews, where eligibility was assessed by two independent reviewers (A.B and G.M.). Articles were first screened independently (A.B. and G.M.) on the basis of title and abstract. A subsequent full-text review was performed by the same reviewers to determine inclusion. All disagreements occurring at the title, abstract, and/or full-text review went to a third reviewer (J.P.S.) for a consensus.

A study was included if it met one or more of the following inclusion criteria: (1) the study evaluated an FC, FL, FIT, neopterin, MMP-9, myeloperoxidase, FCLN-2, CHI3L1, PMN-e, microRNA, S100A12, or alpha-1 antitrypsin for discriminating endoscopically active versus inactive Crohn’s disease in adults (age > 18 years); (2) an endoscopic scoring system or endoscopic description of activity was documented as a reference standard to assess inflammatory activity; (3) the study provided sufficient details to calculate true-positive (TN), false-positive (FP), false-negative (FN), and true-negative (TN) results to (re)construct a two-by-two table.

### 2.3. Data Extraction and Quality Assessment

Two reviewers (A.B. and G.M.) independently retrieved specific data from each full-text article using a standard data extraction form, which included the author, year of publication, nation, method of endoscopic examination (ileocolonoscopy versus balloon assisted enteroscopy), type of faecal biomarker, and endoscopic cut-off values used for each biomarker to discriminate active versus inactive endoscopic CD. The published values for TP, FP, TN, and FN were identified and used to construct a 2 × 2 contingency table. Disagreements between the two reviewers were resolved by consensus. If consensus between the two reviewers could not be reached, a third investigator (J.P.S.) was referred for consensus.

The methodological quality of the included studies was evaluated using the Quality Assessment of Diagnostic Accuracy Studies-2 (QUADAS) criteria [18].

### 2.4. Statistical Analysis

Descriptive statistics were calculated based on the sensitivity, specificity, and false-positive rate of the primary studies, their positive, and negative likelihood ratios (LR+, LR−) and their diagnostic odds ratios (DOR). χ^2^ tests were performed to assess the heterogeneity of sensitivities and specificities. Crosshair plots were performed, and summary receiver operating characteristic (sROC) curves were produced to graphically demonstrate sensitivity and specificity.

Univariate diagnostic odds ratios and negative likelihood ratios were calculated using a fixed effects model and the Mantel–Haensz method with the i^2^ statistic applied to assess heterogeneity. A bivariate model was used to generate pooled sensitivity and specificities. However, as illustrated by Glas et al., these paired indicators are not valid for comparing multiple diagnostic tests, especially if one test was not superior to the other across both indicators [19]. In addition, these parameters cannot be analysed in the traditional framework of network meta-analysis or indirect comparison models. We therefore performed a bivariate analysis of the diagnostic odds ratio (DOR) which is a single global indicator for diagnostic test performance and is defined as the ratio of the odds of positive test results in subjects with the disease, compared to the odds in those without the disease. This bivariate model applied logit transformation of pairs of sensitivity and specificity to provide pooled sensitivity and specificity estimates and to derive the area under the curve analyses.

If 2 × 2 data were reported at multiple thresholds, data related to the optimal cut-off value were extracted. Global statistical heterogeneity was assessed across all comparisons using the i2 measure with the meta-statistical package [20]. Heterogeneity was defined as the following: 0% to 40%, might not be important; 30% to 60%, may represent moderate heterogeneity; 50% to 90%, may represent substantial heterogeneity; and 75% to 100%, considerable heterogeneity [21].

The meta-analysis was performed using the statistical package meta in R (version 4.1.3, R Core Team, The R Foundation, Vienna, Austria). The results were reported according to the PRISMA extension statement for meta-analyses [22]. Finally, comparison-adjusted funnel plots were generated to evaluate publication bias and small-study bias, where sufficient studies (≥10) existed [23].

### 2.5. Deviations from Protocol

In the prespecified protocol for this review [17], the target population was described as patients with IBD. Given the differences in disease patterns and distribution, we focused solely on patients with CD, with a future scope for another review with ulcerative colitis as the target population. In addition, it was originally proposed that the reference standard for diagnosis should include radiological and histological endpoints. However, given that cross-sectional imaging treatment targets and histological treatment targets remained poorly defined at the time of writing, this review was focused on ileocolonoscopy or balloon-assisted endoscopy assessment, which is the current gold standard of CD assessment, as the reference standard. A subanalysis of faecal biomarkers in determining mucosal healing in CD utilising endoscopic descriptions and/or definitions was also performed (where possible) or was otherwise reported with original sensitivities and specificities. In addition, to avoid the risk of duplicate patient inclusion, where studies from the same author in the same year were found, this analysis included the study with the larger number of patients.

Finally, the review also planned to determine the accuracy of faecal biomarkers based on IBD location. However, given the limited number of location-specific studies that were performed across multiple different modalities (e.g., ileocolonoscopy, radiology, and capsule endoscopy), this was not undertaken because the accuracy of faecal biomarker assessment was degraded using the variable application of multiple modalities as the reference standard.

## 3. Results

The initial literature search identified a total of 2382 reports; of these, 2117 were excluded based on their titles and abstracts. A review of the full text of the remaining 266 articles led to an additional 240 being excluded due to failure to meet the inclusion criteria, lacking the data required to create a contingency table, or due to the same author publishing twice within the same year. A further seven studies were identified through reference checking of the included studies. Hence, 33 eligible studies were included (see Figure 1: PRISMA flow chart).

### 3.1. Study Characteristics

The 33 eligible studies included a population of CD patients who undertook faecal biomarker assessment with FC, FL, CHI3L1, MMP-9, FCLN-2, FIT, PMN-e, neopterrin, and/or myeloperoxidase against endoscopic assessment as the reference standard. Thirty-two studies assessed FC, four studies assessed FL, two studies assessed FCLN-2 and FIT, and one study assessed each of PMN-e, MMP-9, neopterrin, CHI3L1, and myeloperoxidase, respectively. A total of 2511 and 230 patients with CD were included in the FC and FL analyses, discriminating between active and inactive endoscopic disease. A total of 1086 patients were included in a separate subanalysis evaluating the capacity of FC to discriminate endoscopic mucosal healing. Endoscopic definitions of CD activity were via the simplified endoscopy score for CD (SES-CD), Crohn’s disease endoscopic index of severity (CDEIS), presence of endoscopic ulcers, and one study used an internally developed author’s score. SES-CD scores to discriminate between endoscopic CD activity and mucosal healing ranged between 0 and 3, and CDEIS between 0 and 6. Han et al. utilised a partial SES-CD whereby the presence of stenosis was excluded from the original SES-CD in determining endoscopic activity [24]. Iwamoto et al. utilised an extended SES-CD whereby CD activity from the proximal ileum and jejunum were included in determining endoscopic activity [25].

### 3.2. Study Quality

The 33 included studies underwent quality assessment with the QUADAS-2 criteria for diagnostic studies, with a summary presented in Figure 2. Twenty-three studies randomly and/or consecutively enrolled patients with CD, whereas the remaining studies were either unclear about or engendered bias via their method selection of patients. Moreover, in 15 studies it was unclear whether blinding of the reference standard was performed, and in 24 studies it was unclear whether blinding of the index test was performed. Overall, concerns regarding the applicability of the included studies across all domains were minimal.

### 3.3. Faecal Calprotectin

There were 32 studies that were eligible for assessment of faecal calprotectin [15,16,24,25,26,27,28,29,30,31,32,33,34,35,36,37,38,39,40,41,42,43,44,45,46,47,48,49,50,51,52,53]. Faecal calprotectin assays were performed via a lateral flow method in six studies, enzyme-linked immunosorbent assay (ELISA) in 25 studies, and the method was not described in one study. Multiple different proprietary faecal calprotectin kits were used, including Calprest (Eurospital Spa, Trieste, Italy), Bühlmann Calprotectin ELISA kit (Bühlmann, Schönenbuch, Switzerland), CALPRO ELISA (Calpro AS, Lysaker, Norway), EliA Calprotectin 2 (Thermo Fisher Scientific, Tokyo, Japan), Quantum Blue test (Buhlmann Laboratories, Schönenbuch, Switzerland), PhiCal (Immundiagnostik AG, Bensheim, Germany), RIDASCREEN^®^ CALPROTECTIN (R-Biopharm AG, Darmstadt, Germany), and PhiCal (Genova Diagnostics Laboratories, Asheville, NC, USA). Definitions of endoscopic activity varied within the included studies, with nine studies applying an SES-CD > 2, five using SES-CD > 3, three using the presence of endoscopic ulcers, two using a CDEIS ≥ 3, and one using a CDEIS > 3. The remaining definitions (extended SES-CD ≥ 1, partial SES-CD ≥ 1, and an internally developed authors’ score) were each used in one study, respectively. Additionally, definitions of mucosal healing varied within the included studies, with three studies applying an SES-CD ≤ 2, three using an SES-CD = 0, and one each using a CDEIS ≤ 3, pSES-CD = 0, and the absence of endoscopic ulcers and/or inflammation.

#### 3.3.1. Detection of Endoscopic Activity

Within the included 25 studies (Table 1), FC sensitivities in discriminating endoscopically active CD ranged from 52 to 97% with a specificity ranging from 45 to 98%. The highest sensitivity was reported by Chen et al. (97%), with the highest specificity reported by Lobaton et al. (98%) [30,39]. The DOR was 13.93 (95% CI 10.89–17.81) with an i^2^ value of 1.34%, suggesting an insignificant heterogeneity and a negative likelihood ratio of 0.27 (95% CI 0.22–0.33) (Figure 3 and Figure 4). The Spearman’s correlation between sensitivity and false positive rate was 0.52. Using bivariate analysis, the pooled sensitivity was 81% (95% CI, 77–84%) with a specificity of 74% (95% CI, 70–80%) and an AUC of 0.85. On assessment using Deek’s funnel plot, there was no evidence of publication bias (*p* = 0.41).

#### 3.3.2. Prediction of Mucosal Healing

Within the nine included studies (Table 2), FC sensitivities in mucosal healing in CD ranged from 75 to 95% with a specificity ranging from 53 to 85%. The highest sensitivity was reported by Vazquez-Moron et al. (95%), and the highest specificity was reported by Castiglione and Cannatelli et al. (85%) [47,53,54]. The DOR was 18.17 (95% CI [11.08–29.82] with an i^2^ value of 0% implying no significant heterogeneity and a negative likelihood ratio of 0.19 [0.14–0.26]) (Figure 5 and Figure 6). The Spearman’s correlation between sensitivity and false positive rate was 0.48. Using bivariate analysis, the pooled sensitivity was 88% (84–90), specificity was 72% (64–79), and AUC was 0.88. There were not enough studies to assess for publication bias.

### 3.4. Faecal Lactoferrin

Lactoferrin is a component of polymorphonuclear neutrophils, which are implicated in the acute inflammatory response as seen in CD [35]. Four studies were eligible for the assessment of sensitivity and specificity for faecal lactoferrin in discriminating endoscopic activity (see Table 3) [31,34,35,43]. All lactoferrin assays were performed via ELISA, with four studies using the IBD-SCAN^®^ (Techlab, Blacksburg, VA, USA) kit and one study using the IBD-CHEK^®^ (Techlab, Blacksburg, VA, USA) kit [31,34,35,43]. Definitions of endoscopic activity varied within the included studies, with one study each using a CDEIS > 3, CDEIS > 2, SES-CD > 3, and an internally developed authors’ score to discriminate between active and inactive endoscopic CD [31,34,35,43].

Figure 7 and Figure 8 show the forest plots for the negative likelihood ratio and DOR of FL in discriminating endoscopically active and inactive CD. Sensitivities ranged from 66 to 81%, with the highest sensitivity reported as 81% by Langhorst et al. [35]. The specificity ranged from 59 to 91%, with the highest specificity of 91% reported by Sipponen et al. [43]. The negative likelihood ratio was 0.34 (95% CI [0.26–0.45] and the DOR was 13.42 (95% CI: 5.74–31.32) with an I^2^ value of 0%, suggesting no heterogeneity. The Spearman’s correlation between sensitivity and false positivity was 0.89. Using bivariate analysis, the pooled sensitivity was 75% (65–83) and the pooled specificity was 80% (57–92), with an AUC of 0.81. There were not enough studies to assess for publication bias.

### 3.5. Other Biomarkers

Faecal Lipocalin-2

FCLN-2 is a glycoprotein produced by intestinal epithelial cells and released into the gut lumen in response to proinflammatory stimuli, as occurs in CD [27,46]. FCLN-2 assays are performed via ELISA and have been described in two studies as potential discriminant faecal biomarkers of endoscopically active CD [27,46]. In a cohort of 54 patients with CD, Buisson et al. demonstrated an optimal cut-off level of 6700 ng/g for discriminating between the absence and presence of endoscopic ulcers, with a sensitivity and specificity of 86% and 46%, respectively [27]. More recently, in a cohort of 72 patients with CD, Zollner et al. demonstrated an optimal cut-off of 0.56 µg/g for discriminating endoscopically active and inactive CD (defined as an SES-CD > 3 for active CD), with a sensitivity and specificity of 91% and 77%, respectively [46]. The discrepancy in optimal cut-offs between the two studies may be explained by the different endoscopic standards applied (i.e., ulceration versus defined SES-CD score).
Faecal metalloprotease 9

Matrix metalloproteases are known to activate or degrade a variety of substrates within an immune response [27]. MMP-9 is expressed in inflamed intestinal mucosa by macrophages and neutrophils and released into faeces during active CD [27]. MMP-9 assays are performed via ELISA and have been described in one study as a potential discriminant faecal biomarker of endoscopically active CD. In a cohort of 54 patients with CD, Buisson et al. demonstrated an optimal cut-off level of 350 ng/g for discriminating between the absence and presence of endoscopic ulcers, with a sensitivity and specificity of 90% and 64%, respectively [27].
Neopterin

Neopterin is produced and released primarily from macrophages in response to stimulation from activated T-cells, as occurs in CD [38]. Neopterin assays are performed via ELISA and have been described in one study as a potential faecal biomarker for discriminating endoscopically active CD [38]. In a cohort of 78 patients with CD, Nancey et al. demonstrated sensitivity and specificity of 74% and 73%, utilising a neopterin cut-off value of 200 pmol/g for discriminating endoscopically active CD (defined as an SES-CD > 3) [38].
Faecal myeloperoxidase

Myeloperoxidase is an enzyme present within neutrophils with a putative role in inflammatory tissue damage [44]. Its presence within faeces is detectable via ELISA, with one study demonstrating its potential as a biomarker for discriminating endoscopically active CD [44]. In a cohort of 100 patients with CD, Swaminathan et al. demonstrated an optimal faecal myeloperoxidase cut-off level of 10.25 µg/g for discriminating endoscopically active CD (defined as an SES-CD > 3) [44]. At this cut-off, faecal myeloperoxidase demonstrated a sensitivity and specificity of 63 and 69%, respectively [44].
Faecal chitinase 3-like 1

Chitinase 3-like 1(CH3L1) is expressed in a variety of cells such as macrophages and neutrophils and is upregulated in the colonic epithelial cells and lamina propria macrophages of inflamed mucosa in CD [52]. Its presence within faeces is detectable via ELISA, with one study demonstrating its potential as a biomarker for discriminating endoscopically active CD [52]. In a study of 54 patients with CD, Buisson et al. determined an optimal faecal CH3L1 cut-off level of 15 ng/g for discriminating endoscopic ulceration in CD [52]. At this cut-off level, faecal CH3L1 demonstrated a sensitivity and specificity of 100% and 64%, respectively [52].
PMN-e

Polymorphonuclear neutrophil elastase (PMN-e) is an enzyme stored in polymorphonuclear neutrophils and released during the activation of these cells as a mediator of inflammation [35]. Its presence within faeces is detectable via ELISA, with one study demonstrating its potential as a biomarker capable of discriminating endoscopically active CD [35]. In a study of 43 patients with CD, Langhorst et al. determined an optimal faecal PMN-e cut-off level of 0.062 µg/mL for discriminating endoscopically active CD (using an internally developed endoscopic score by the authors as the reference standard) [35]. At this cut-off level, faecal PMN-E demonstrated a sensitivity and specificity of 82% and 70%, respectively [35].
FIT

FIT quantitatively measures faecal haemoglobin concentrations and forms the backbone of colon cancer surveillance programs worldwide [48]. Quantitative measurements are widely performed using the proprietary test from the Eiken Chemical Co. (Tokyo, Japan). In a cohort of 69 patients, Iwamoto et al. determined an FIT cut-off of 13 ng/mL to have a sensitivity and specificity of 73% and 71% in discriminating endoscopically active CD (defined as an eSES-CD ≥ 1) [25]. Regarding mucosal healing, in a cohort of 71 patients, Inokuchi et al. determined that a FIT cut-off level of 52 ng/mL was predictive of mucosal healing in CD (defined as an SES-CD = 0), with a sensitivity and specificity of 96% and 48%, respectively [48]. Other biomarkers used to detect endoscopic activity in CD are summarized in Table 4.

## 4. Discussion

With regular objective assessment now recommended by international guidelines, there is a growing imperative for cheaper, rapidly accessible yet reliable tests for assessing disease activity in CD. Clinical activity scores, such as the Harvey-Bradshaw index or CD activity index (CDAI), are predominantly symptom-based, and therefore subjective and prone to providing falsely reassuring results [32]. Moreover, biochemical tests, such as C-reactive protein, remain nonspecific to IBD and are subject to confounding in multiple clinical settings. Whilst ileocolonoscopy and cross-sectional imaging remain important options due to their informational capacity, neither are cheap nor necessarily rapidly available. Furthermore, both require significant patient participation, preparation, and risks that might be deemed unacceptable for repetitive testing over short six-monthly assessment intervals, as recommended by STRIDE-2 guidelines. Thus, faecal biomarkers play a significant role in the routine, serial monitoring of IBD patients.

This meta-analysis strengthens the existing knowledge regarding FC as a capable biomarker in discriminating between active/inactive endoscopic disease and mucosal healing and provides context for FC amongst other more novel faecal biomarkers and their respective performance in discriminating endoscopically active versus inactive CD. This meta-analysis reaffirmed that both FC and FL are highly sensitive and specific for discriminating active versus inactive endoscopic disease in CD, with the inclusion of a number of more recent studies since the previous meta-analysis was performed many years ago [55]. In this meta-analysis, FC demonstrated a pooled sensitivity of 81% and specificity of 74% (AuROC and DOR of 0.85 and 13.93, respectively), implying that FC retains good diagnostic capacity across all disease subtypes and locations of the heterogeneous entity that is CD. These pooled data are similar to those previously reported for FC in the literature [55]. Moreover, FC exhibits an excellent capacity for prediction of mucosal healing with a summary DOR of 18.17 (95% CI 11.08–29.82) in this study.

Many FC manufacturers recommend 50 µg/g as a cut-off value for representing active intestinal inflammation. However, in this meta-analysis, 15 of 25 (60%) studies reported an FC > 200 µg/g as their optimal cut-off when discriminating between endoscopically active and inactive disease, albeit with heterogeneous definitions of endoscopically active CD. Comparatively, with respect to mucosal healing, only two studies derived a FC > 200 µg/g as optimal and most (5/9) applied a normal endoscopy (SES-CD = 0 and/or no ulcers/inflammation) as their definition of mucosal healing. Hence, the capacity of a faecal biomarker to surrogately represent mucosal healing may be a more appropriate ‘harder’ endpoint by which to determine its accuracy and utility when correlating to endoscopy as a reference standard. While determining the optimal cut-off for FC in detecting endoscopically active or mucosal healing in CD was beyond the scope of this analysis, it appears that a lower calprotectin cut-off at a higher sensitivity is demonstrated in studies addressing mucosal healing compared to endoscopic disease activity in CD. Ultimately, further studies examining newer calprotectin assays and similar definitions of endoscopic activity (and/or histological activity as some recent studies have explored) are needed.

Within our meta-analysis, FL demonstrated a pooled sensitivity, specificity, AuROC, and DOR of 75%, 80%, 0.81, and 13.41, respectively. Compared to FC, the performance of FL was similar, yet the role of FL, for unknown reasons beyond the scope of this study, has not been incorporated into widespread clinical practice. The diagnostic performance of alternative faecal biomarkers was included in the study, yet most were limited to single studies. Thus, it remains difficult to draw meaningful conclusions from these biomarkers regarding their capacity to discriminate between active and inactive endoscopic CD or mucosal healing. Further studies of biomarkers beyond FC and FL are clearly needed to determine and establish their potential role in CD assessment, especially concerning whether they offer advantages over FC in terms of cost, accuracy, and/or convenience. In addition, combinations of low-cost faecal biomarkers should be explored in an effort to enhance diagnostic performance.

This meta-analysis was subject to several weaknesses and limitations. The data were drawn from predominately retrospective studies with their inherent biases. Moreover, our endoscopic reference standard was variably applied via multiple, different endoscopic scores and cut-off levels therein used to discriminate CD activity which may have impacted the observed results. In addition, FC and FL assays were performed via different methods (i.e., lateral flow and ELISA) which may not be of equivalent accuracy.

## 5. Conclusions

Based on this rigorous meta-analysis, FC and FL were both shown to perform robustly in discriminating between active and inactive CD. Both are relatively cheap, rapid, and reliable tests that are therefore amenable to serial testing at frequent intervals to confidently monitor CD activity. Newer and novel faecal biomarkers have been proposed but require further evaluation before any could be considered as a replacement, let alone superior, to FC in routine clinical practice. There is clearly an unmet need to incisively examine the capacity of faecal biomarkers to assess CD activity according to disease location, extent, and specific subtypes. Moreover, in this context, the application of multiple biomarkers (faecal, serum, and/or other) in a combined diagnostics matrix, perhaps harnessing artificial intelligence-based algorithms, may provide even greater diagnostic and predictive power. Faecal biomarkers are here to stay and, with further research, are likely to continue to increasingly dominate the diagnostics landscape in routine management of CD.

## Figures and Tables

**Figure 1 biomedicines-11-01408-f001:**
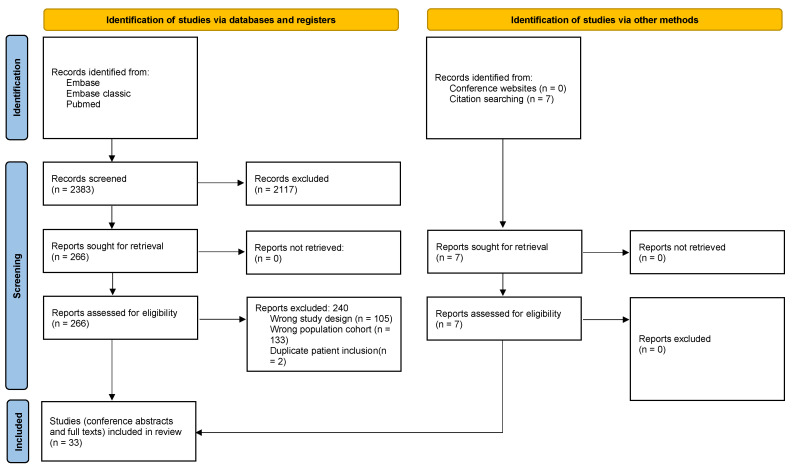
PRISMA flow chart.

**Figure 2 biomedicines-11-01408-f002:**
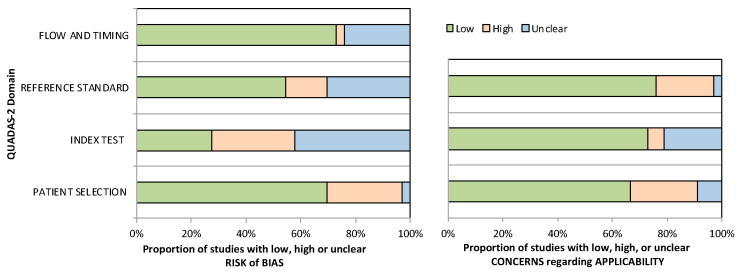
Graphical display of QUADAS-2 results.

**Figure 3 biomedicines-11-01408-f003:**
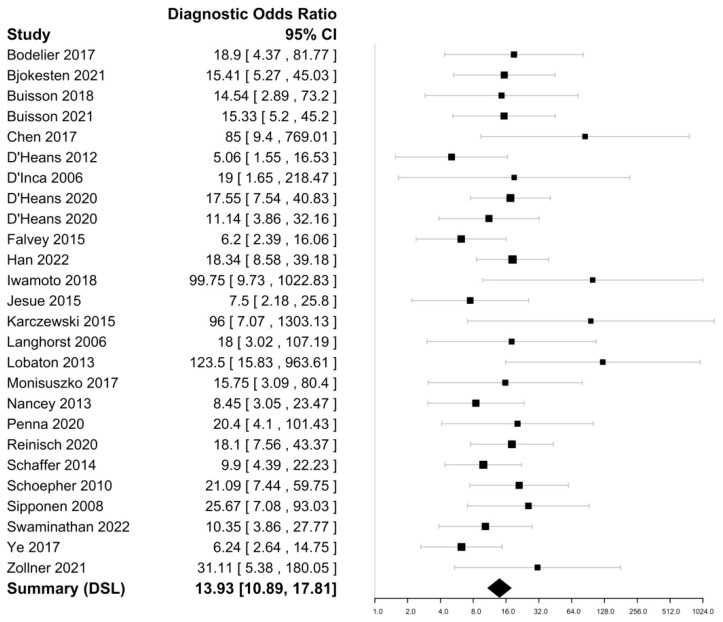
Forest plots of diagnostic odds ratio of faecal calprotectin for CD endoscopic activity assessment [15,16,24,25,26,27,28,29,30,31,32,33,34,35,36,37,38,39,40,41,42,43,44,45,46].

**Figure 4 biomedicines-11-01408-f004:**
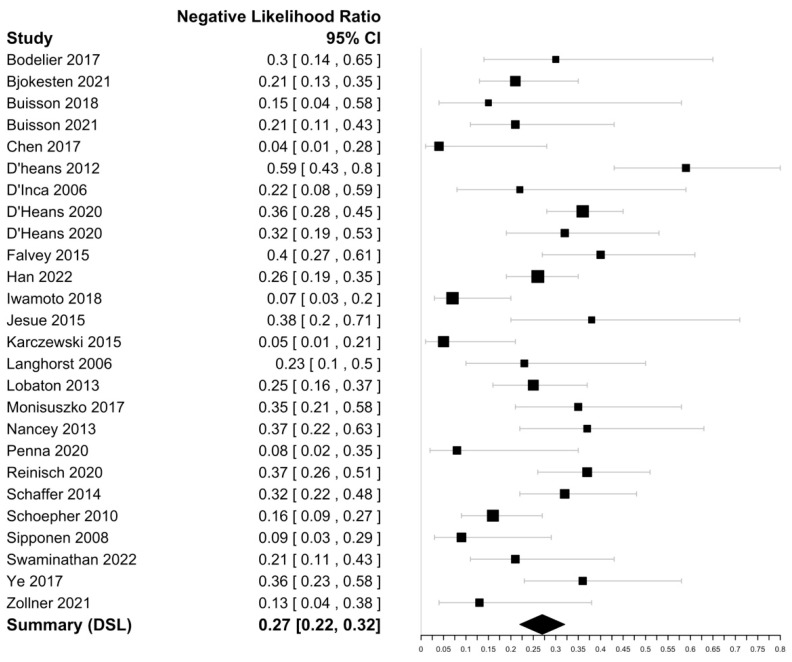
Forest plots of negative likelihood ratio estimate of faecal calprotectin for CD endoscopic activity assessment [15,16,24,25,27,28,29,30,31,32,33,34,35,36,37,38,39,40,41,42,43,44,45,46].

**Figure 5 biomedicines-11-01408-f005:**
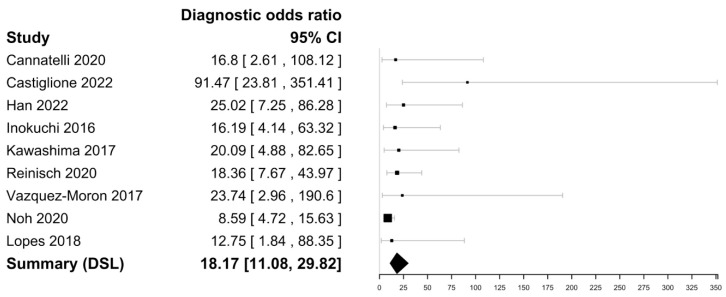
Forest plots of diagnostic odds ratio of faecal calprotectin for CD mucosal healing assessment [24,40,47,48,49,50,51,53,54].

**Figure 6 biomedicines-11-01408-f006:**
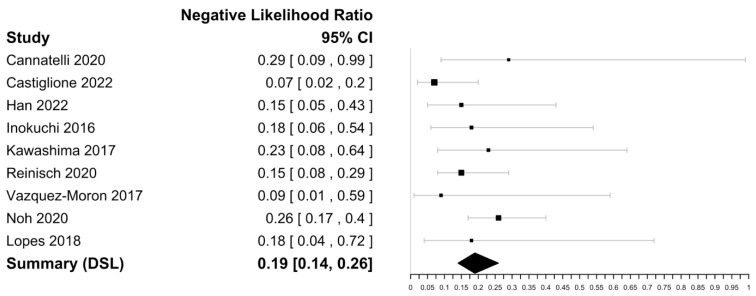
Forest plots of negative likelihood ratio estimate of faecal calprotectin for CD mucosal healing assessment [24,40,47,48,49,50,51,53,54].

**Figure 7 biomedicines-11-01408-f007:**
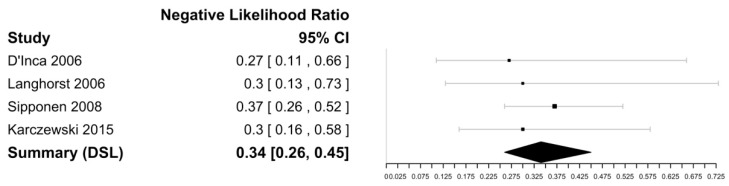
Forest plot for the negative likelihood ratio of FL in discriminating endoscopically active and inactive CD [33,36,38,45].

**Figure 8 biomedicines-11-01408-f008:**
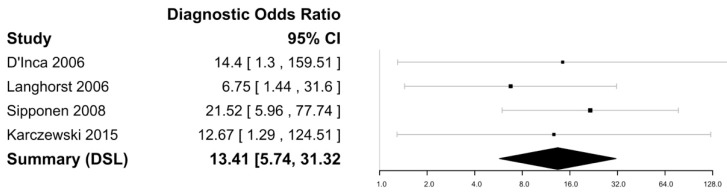
Forest plot for the diagnostic odds ratio of FL in discriminating endoscopically active and inactive CD [31,34,35,43].

**Table 1 biomedicines-11-01408-t001:** Summary of FC studies assessing endoscopic activity in Crohn’s Disease(CD).

Study (Name, Year)	Nation of Origin	CD Patients (n=)	Definition of Endoscopic Activity (Inactive/Active)	Optimal FC Cut-off	Sensitivity (95% CI)	Specificity (95% CI)
Björkesten 2021 [15]	Finland	126	SES-CD ≤ 2/SES-CD > 2	94 µg/g	84% (76–90)	74% (54–87)
Bodelier 2017 [26]	Netherlands	50	SES-CD ≤ 3/SES-CD > 3	250 µg/g	74% (51–88)	87% (71–95)
Buisson 2021 [16]	France	83	Absence of ulcers/ presence of ulcers	250 µg/g	84% (70–91)	74% (58–85)
Buisson 2018 [27]	France	54	Absence of ulcers/presence of ulcers	250 µg/g	91% (73–98)	58% (41–73)
Chen 2017 [28]	China	56	SES-CD ≤ 2/SES-CD > 2	250 µg/g	97% (85–99)	71% (50–86)
D’Haens 2012 [29]	Netherlands	87	Absence of ulcers/presence of ulcers	250 µg/g	52% (40–63)	83% (63–93)
D’Haens 2020 * [30]	Netherlands	247 81	SES-CD ≤ 2/SES-CD > 2 SES-CD ≤ 2/SES-CD > 2	250 µg/g 50 µg/g	68% (61–75) 75% (61–85)	88% (78–94) 78% (62–89)
D’Inca 2006 [31]	Italy	31	SES-CD ≤ 3/SES-CD > 3	80 µg/g	83% (63–93)	80% (38–96)
Falvey 2015 [32]	New Zealand	108	SES-CD ≤ 2/SES-CD > 2	125 µg/g	71% (60–80)	71% (53–84)
Han 2022 [24]	China	254	pSES-CD = 0/pSES-CD ≥ 1	156 µg/g	78% (72–83)	83% (72–90)
Iwamoto 2018 [25]	Japan	69	eSES-CD = 0/eSES-CD ≥ 1	92 mg/kg	93% (84–97)	88% (53–91)
Jesue 2018 [33]	Spain	52	SES-CD = 0/SES-CD ≥ 1	54 µg/g	71% (53–84)	75% (55–88)
Karczewski 2015 [34]	Poland	55	CDEIS < 3/CDEIS ≥ 3	76 µg/g	96% (87–99)	80% (38–96)
Langhorst 2006 [35]	Germany	43	Authors’ score	48 µg/mL	82% (66–91)	80% (49–94)
Lobaton 2013 [36]	Spain	115	CDEIS < 3/CDEIS ≥ 3	274 µg/g	76% (65–84)	98% (87–99)
Monisuszko 2017 [37]	Poland	57	SES-CD ≤ 3/SES-CD > 3	238.5 µg/g	69% (53–81)	88% (64–97)
Nancey 2013 [38]	France	78	SES-CD ≤ 2/SES-CD > 2	250 µg/g	71% (55–82)	77% (62–87)
Penna 2020 [39]	Italy	80	SES-CD ≤ 2/SES-CD > 2	155 µg/g	96% (87–99%)	45% (28–63)
Reinisch 2020 [40]	Austria	156	CDEIS ≤ 3/CDEIS > 3	250 µg/g	67% (56–76)	90% (81–95)
Schaffer 2014 [41]	Switzerland	136	SES-CD ≤ 3/SES-CD > 3	250 µg/g	75% (65–83%)	76% (63–86)
Schoepher 2010 [42]	Switzerland	122	SES-CD ≤ 3/SES-CD > 3	70 µg/g	89% (81–93)	72% (53–86)
Sipponen 2008 [43]	Finland	116	CDEIS ≤ 3/CDEIS > 3	200 µg/g	94% (84–98)	61% (48–73)
Swaminathan 2022 [44]	New Zealand	100	SES-CD ≤ 2/SES-CD > 2	58 µg/g	87% (76–93)	61% (45–74)
Ye 2017 [45]	China	109	SES-CD ≤ 2/SES-CD > 2	213 µg/g	76% (65–84)	66% (51–79)
Zollner 2021 [46]	Austria	72	SES-CD ≤ 2/SES-CD > 2	78 µg/g	90% (75–97)	77% (50–92)

* D’Haens 2020 had 2 groups measured separately. Abbreviations: SES-CD, simplified endoscopy score for Crohn’s disease; CDEIS, Crohn’s disease endoscopic index of severity; pSES-CD, partial simplified endoscopy score for Crohn’s disease; eSES-CD, extended simplified endoscopy score for Crohn’s disease.

**Table 2 biomedicines-11-01408-t002:** Summary of FC studies assessing mucosal healing in CD.

Study (Name, Year)	Nation of Origin	Definition of Endoscopic Mucosal Healing	Optimal FC Cut-off	Sensitivity (95% CI)	Specificity (95% CI)
Cannatelli 2021 [53]	Italy	SES-CD ≤ 2	96 mcg/g	75% (41–93%)	85% (69–93)
Castiglione 2022 [54]	Italy	SES-CD ≤ 2	94 µg/g	94% (84–98)	85% (74–92)
Han 2022 [24]	China	pSES-CD = 0	117.48 µg/g	89% (72–86)	76% (70–81)
Inokuchi 2016 [48]	Japan	SES-CD = 0	180 µg/g	87% (68–95)	71% (57–82)
Kawashima 2017 [49]	Japan	SES-CD ≤ 2	162.2 µg/g	81% (57–93)	82% (71–90)
Lopes 2018 [50]	Spain	SES = CD = 0	100 µg/g	89% (69–97)	60% (31–83)
Noh 2018 [51]	Korea	No ulcers and/or inflammation	234 µg/g	84% (76–90)	62% (54–69)
Reinisch 2020 [40]	Austria	CDEIS ≤ 3	250 µg/g	90% (81–95)	67% (56–77)
Vazquez-Moron 2017 [47]	Spain	SES-CD ≤ 2	71 µg/g	95% (78–99)	53% (39–66)

Abbreviations: SES-CD, simplified endoscopy score for Crohn’s disease: CDEIS, Crohn’s disease endoscopic index of severity; pSES-CD, partial simplified endoscopy score for Crohn’s disease.

**Table 3 biomedicines-11-01408-t003:** Summary of studies assessing accuracy of FL to CD endoscopic activity.

Study (Name, Year)	Nation of Origin	CD Patients (n=)	Definition of Endoscopic Activity	Optimal FL Cut-off	Sensitivity (95% CI)	Specificity (95% CI)
D’Inca 2006 [31]	Italy	31	SES-CD ≤ 3/SES-CD > 3	0.007 optical density	77% (57–89)	80% (38–96)
Karczewski 2015 [34]	Poland	55	CDEIS ≤ 2/CDEIS > 2	25 µg/g	75% (62–85)	80% (38–96)
Langhorst 2006 [35]	Germany	43	Authors score	7.1 µg/mL	81% (65–91)	59% (32–82)
Sipponen 2008 [43]	Finland	116	CDEIS ≤ 3/CDEIS > 3	10 µg/g	66% (54–76)	91% (77–96)

Abbreviations: SES-CD: simplified endoscopy score for Crohn’s disease, CDEIS: Crohn’s disease endoscopic index of severity.

**Table 4 biomedicines-11-01408-t004:** Summary of other faecal biomarkers used to assess endoscopic activity in CD.

Study (Name, Year)	Nation of Origin	Number of CD Patients	Definition of Endoscopic Activity	Faecal Biomarker	Optimal Faecal Biomarker Cut-off	Sensitivity (95% CI)	Specificity (95% CI)
Buisson 2016 [52]	France	54	Absence of ulcers/Presence of ulcers	Faecal chitinase 3-like 1	15 ng/g	100% (84–100)	64% (41–80)
Buisson 2018 [27]	France	54	Absence of ulcers/Presence of ulcers Absence of ulcers/Presence of ulcers	FCLN-2 Faecal metalloprotease 9	6700 ng/g 350 ng/g	86% (64–97) 90% (64–97)	45% (24–68) 64% (41–80)
Iwamoto 2018 [25]	Japan	69	eSES-CD = 0/eSES-CD ≥ 1	FIT	13 ng/mL	73% (95% CI na)	71% (95% CI na)
Langhorst 2006 [35]	Germany	43	Authors score	PMN-e	0.062 µg/mL	82% (65–93)	70% (35–93)
Nancey 2013 [38]	France	78	SES-CD ≤ 2/SES-CD > 2	Neopterin	200 pmol/g	74% (57–87)	73% (56–85)
Swaminathan 2022 [44]	New Zealand	100	SES-CD ≤ 2/SES-CD > 2	Faecal Myeloperoxidase	10.25 µg/g	63% (50–75)	68% (51–83)
Zollner 2021 [46]	Austria	72	SES-CD ≤ 2/SES-CD > 2	FLCN-2	0.56 µg/g	91% (74–98)	77% (46–95)

Abbreviations: SES-CD, simplified endoscopy score for Crohn’s disease; CDEIS, Crohn’s disease endoscopic index of severity; pSES-CD, partial simplified endoscopy score for Crohn’s disease; eSES-CD, extended simplified endoscopy score for Crohn’s disease; FIT, faecal immunochemical test; FCLN-2, faecal lipocalin-2; PMN-e, polymorphonuclear neutrophil elastase, na = not available.

## Data Availability

Available in body of text and Appendix A.

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
