# Peer review of "The Utility of Faecal Calprotectin, Lactoferrin and Other Faecal Biomarkers in Discriminating Endoscopic Activity in Crohn’s Disease: A Systematic Review and Meta-Analysis"

_biomedicines, 2023, doi:10.3390/biomedicines11051408_

Round 1
Reviewer 1 Report
- This meta-analysis represents an important contribution to the field of inflammatory bowel disease (IBD) research, providing a comprehensive overview of the accuracy of various faecal biomarkers in discriminating endoscopic activity and mucosal healing in Crohn's disease.
- The authors should be commended for their rigorous methodology, including a comprehensive literature search and assessment of study quality using the QUADAS criteria.
- The findings of this meta-analysis are highly informative and will be of great value to clinicians and researchers alike, helping to guide the use of faecal biomarkers in the monitoring and management of Crohn's disease.
- The authors' emphasis on the need for further research in this area is particularly noteworthy, highlighting the potential for novel faecal biomarkers such as faecal lipocalin-2, faecal metalloprotease 9, neopterin, faecal myeloperoxidase, faecal chitinase 3-like 1, and polymorphonuclear neutrophil elastase to improve the accuracy of non-invasive disease monitoring.
- Overall, this meta-analysis provides a valuable synthesis of the existing literature on faecal biomarkers in Crohn's disease, and will undoubtedly serve as a useful reference for future research in this field.
Author Response
Dear Reviewer
We thank you for taking the time to review our submitted manuscript. We hope other readers will also find this a useful source of information for future research.
Regards
Dr. Anuj Bohra
Reviewer 2 Report
The authors performed a meta-analysis of faecal biomarkers consistent with discriminate endoscopic activity in Crohn's disease.
Calprotectin has been a well-known biomarker for some time and was similar in this study.
In addition, lactoferrin was also found to be a promising biomarker.
The following points should be cleared up
For example, Table 1 also lists several pairs of publications by the same author with the same or similar publication year. Does the overlap of cases in these reports not affect the analysis? What is the guarantee of no overlap?
Otherwise, the analysis seems to be problem-free and useful.
Author Response
Dear Reviewer
We thank you for taking the time to review our submitted manuscript. We thank you for bringing to our attention the potential overlap of patients between studies from the same author published within the same year. This refers directly to Sipponen et al studies from 2008 and Ye et al studies from 2017. As an abundance of caution, we have removed the Sipponen et al study titled “Fecal calprotectin, lactoferrin, and endoscopic disease activity in monitoring anti-TNF-alpha therapy for Crohn's disease” published in Inflammatory Bowel Diseases and Ye at al titled ““Fecal calprotectin is a strong predictive marker of relapse in Chinese patients with Crohn’s disease: a two-year prospective study” published in Scandinavian Journal of Gastroenterology. These were picked on the basis that they had lower numbers of included patients.
This reduces the total number of patients included in the faecal calprotectin and faecal lactoferrin to endoscopic activity analysis to 2511(from 2600) in the faecal calprotectin meta analysis and 230(from 260) in the faecal lactoferrin meta analysis. On subsequent re-analysis, no significant changes in other cohorts were observed. Faecal calprotectin was found to have a pooled sensitivity and specificity, diagnostic odds ratio and negative predictive value in discriminating active endoscopic disease (versus inactive) of 81%, 74%, 13.93 and 0.27 respectively. Faecal lactoferrin had a pooled sensitivity and specificity, diagnostic odds ratio, and negative predictive value in discriminating active endoscopic disease of 75%, 80%, 13.41 and 0.34 respectively. The the entire manuscript submission has been updated to reflect the minor updated results with tracked changes.
We thank you for raising this query with us and providing the authors to opportunity to reflect on our analysis and further improve the quality and accuracy of our manuscript.
Regards
Dr. Anuj Bohra
Reviewer 3 Report
Review for the manuscript “Faecal biomarkers used to discriminate endoscopic activity in Crohn’s Disease: A meta-analysis” submitted to Biomedicines.
Dear Editor, thank you for the invitation to review this manuscript. After evaluation, I suggest some modifications before it can be considered for publication.
Overall comments: This is an interesting study that aimed to perform a meta-analysis to investigate the role of fecal biomarkers (lactoferrin and calprotectin in discriminating endoscopic activity in Crohn’s Disease.
ABSTRACT
Please check the number of words. I think it exceeds the allowed by MDPI.
In lines 17-19, we can read: “However, there are several potential faecal biomarkers 17 described in the literature. We performed a meta-analysis to determine the accuracy of FC and other fecal biomarkers in discriminating endoscopic activity and mucosal healing in CD.” Please cite which are these other fecal markers.
Later in the abstract, we can find “Other novel biomarkers insufficient published data 30 to derive accuracy statistics.” So the investigated biomarkers were calprotectin and lactoferrin. So, I suggest modifying the title to “Faecal calprotectin and lactoferrin in discriminating endoscopic activity in Crohn’s Disease: A meta-analysis.”
INTRODUCTION
I suggest including newer references to introduce Crohn´s Disease (please, include references published in 2022-2023). A plethora of very nice studies published in 2023 can be found in PUBMED.
In this section and along with all the text, please, include a space between the sentences and the parenthesis of the cited references.
In lines 63-65, we can read “Furthermore, a number of 63 other faecal biomarkers of promising utility for discriminating endoscopic CD activity 64 have been described in the literature with the potential to be incorporated more widely 65 into routine clinical practice.” Please, describe the other fecal biomarkers.
OBJECTIVES:
In the end of the Introduction section we see “Therefore, this meta-analysis was conducted to assess and 66 re-affirm the accuracy of FC and examine other studied faecal biomarkers in discriminat-67 ing endoscopic activity in CD.”. I suggest modifying for “Therefore, this meta-analysis was conducted to assess the accuracy of FC and lactoferrin in discriminating endoscopic activity in CD.”
METHODS
In lines 88-91 we see: “A study was included if it met one or more of the following inclusion criteria: (1) the 89 study evaluated a faecal biomarker (FC), faecal lactoferrin(FL), faecal immunochemical 90 test(FIT), neopterin, metalloprotease-9(MMP-9), myeloperoxidase, faecal lipocalin-91 2(FCLN-2), chitinase 3-like 1(CHI3L1), polymorphonuclear neutrophil elastase(PMN-e)…”. Please, include a space between the word before the ():
“A study was included if it met one or more of the following inclusion criteria: (1) the study evaluated a faecal biomarker (FC), faecal lactoferrin (FL), faecal immunochemical test (FIT), neopterin, metalloprotease-9 (MMP-9), myeloperoxidase, faecal lipocalin- 2 (FCLN-2), chitinase 3-like 1 (CHI3L1), polymorphonuclear neutrophil elastase (PMN-e)…”
In line 166, please, include the reference for PRISMA flowchart (Moher, 2009 or Page 2021).
In Figure 2, we see “Figure 2. Graphical display of QUDAS-2 results.” Would it be “Figure 2. Graphical display of QUADAS-2 results.”?
RESULTS AND DISCUSSION
I do not think it is necessary to keep Figures 5, 8, and 11.
I also suggest the authors consider removing (or reducing) section 3.5 (found in lines 296-363) and Table 4.
CONCLUSION
This section is adequately described.
REFERENCES
As pointed out above, I suggest including newer references in the Introduction section and along with the text.
Author Response
Dear Reviewer
We thank you for taking the time to review our submitted manuscript. We thank you for your valuable feedback on our submitted manuscript and appreciate the opportunity to correct and improve our manuscript based upon your feedback.
With regards to your specific comments:
ABSTRACT
Please check the number of words. I think it exceeds the allowed by MDPI.
In lines 17-19, we can read: “However, there are several potential faecal biomarkers 17 described in the literature. We performed a meta-analysis to determine the accuracy of FC and other fecal biomarkers in discriminating endoscopic activity and mucosal healing in CD.” Please cite which are these other fecal markers.
Later in the abstract, we can find “Other novel biomarkers insufficient published data 30 to derive accuracy statistics.” So the investigated biomarkers were calprotectin and lactoferrin. So, I suggest modifying the title to “Faecal calprotectin and lactoferrin in discriminating endoscopic activity in Crohn’s Disease: A meta-analysis.”
Response: We thank the reviewer for highlighting the word count requirement for the manuscript. We have reduced the word count to 213 words and have emailed the editorial office to ensure this falls within the allowed limits. In light of the word count we are limited and unable to add in the other novel faecal biomarkers assessed in the abstract introduction section. With regards to the title, our study aim was to address all faecal biomarkers though faecal calprotectin and lactoferrin provided enough studies to perform an analysis. The study still covers and describes other assessed faecal biomarkers which makes this unique compared to other studies previously described in the literature which focus only on a single biomarker. Therefore, we have changed the title to “The utility of faecal calprotectin, lactoferrin and other faecal biomarkers in discriminating endoscopic activity in Crohn’s Disease: A systematic review and meta-analysis.” Which more accurately reflect the body of work conducted in performing this analysis.
INTRODUCTION
I suggest including newer references to introduce Crohn´s Disease (please, include references published in 2022-2023). A plethora of very nice studies published in 2023 can be found in PUBMED.
In this section and along with all the text, please, include a space between the sentences and the parenthesis of the cited references.
In lines 63-65, we can read “Furthermore, a number of 63 other faecal biomarkers of promising utility for discriminating endoscopic CD activity 64 have been described in the literature with the potential to be incorporated more widely 65 into routine clinical practice.” Please, describe the other fecal biomarkers.
Response: We thank the reviewer for their valuable feedback. A number of more recent references to introduce crohn’s disease have been cited including:
Agrawal M, Christensen HS, Bøgsted M, Colombel JF, Jess T, Allin KH. The Rising Burden of Inflammatory Bowel Disease in Denmark Over Two Decades: A Nationwide Cohort Study. Gastroenterology. 2022;163(6):1547-54.e5.
Lee JW, Eun CS. Inflammatory bowel disease in Korea: epidemiology and pathophysiology. Korean J Intern Med. 2022;37(5):885-94.
Cho CW, You MW, Oh CH, Lee CK, Moon SK. Long-term Disease Course of Crohn's Disease: Changes in Disease Location, Phenotype, Activities, and Predictive Factors. Gut Liver. 2022;16(2):157-70
Kumar A, Cole A, Segal J, Smith P, Limdi JK. A review of the therapeutic management of Crohn's disease. Therap Adv Gastroenterol. 2022;15:17562848221078456
The entire manuscript has been reviewed and layout changed have been addressed including the requested spaces between parenthesis and cited references throughout article. We have now included the name of all faecal biomarkers included in this review with the section now reading “Furthermore, a number of other faecal biomarkers of promising utility for discriminating endoscopic CD activity have been described in the literature with the potential to be incorporated more widely into routine clinical practice including faecal lactoferrin (FL), faecal immunochemical test (FIT), neopterin, metalloprotease-9 (MMP-9), myeloperoxidase, faecal lipocalin-2 (FCLN-2), chitinase 3-like 1 (CHI3L1), polymorphonuclear neutrophil elastase (PMN-e), microRNA, S100A12 and alpha-1 antitrypsin”
OBJECTIVES:
In the end of the Introduction section we see “Therefore, this meta-analysis was conducted to assess and 66 re-affirm the accuracy of FC and examine other studied faecal biomarkers in discriminat-67 ing endoscopic activity in CD.”. I suggest modifying for “Therefore, this meta-analysis was conducted to assess the accuracy of FC and lactoferrin in discriminating endoscopic activity in CD.”
Response: We thank the reviewer for highlighting this issue. As per the response to the abstract section the manuscript still covers and describes other assessed faecal biomarkers thus we have changed the title to “The utility of faecal calprotectin, lactoferrin and other faecal biomarkers in discriminating endoscopic activity in Crohn’s Disease: A systematic review and meta-analysis.” Which more accurately reflect the body of work conducted in performing this analysis.
METHODS
In lines 88-91 we see: “A study was included if it met one or more of the following inclusion criteria: (1) the 89 study evaluated a faecal biomarker (FC), faecal lactoferrin(FL), faecal immunochemical 90 test(FIT), neopterin, metalloprotease-9(MMP-9), myeloperoxidase, faecal lipocalin-91 2(FCLN-2), chitinase 3-like 1(CHI3L1), polymorphonuclear neutrophil elastase(PMN-e)…”. Please, include a space between the word before the ():
“A study was included if it met one or more of the following inclusion criteria: (1) the study evaluated a faecal biomarker (FC), faecal lactoferrin (FL), faecal immunochemical test (FIT), neopterin, metalloprotease-9 (MMP-9), myeloperoxidase, faecal lipocalin- 2 (FCLN-2), chitinase 3-like 1 (CHI3L1), polymorphonuclear neutrophil elastase (PMN-e)…”
In line 166, please, include the reference for PRISMA flowchart (Moher, 2009 or Page 2021).
In Figure 2, we see “Figure 2. Graphical display of QUDAS-2 results.” Would it be “Figure 2. Graphical display of QUADAS-2 results.”?
Response: We thank the reviewer for their valuable feedback. All layout issues space before the (brackets) have now been amended with further corrections to be made by the MDPI layout team in accordance with the manuscript requirement. A citation for Moher 2009 has been included within the methods section. We also thank the reviewer for highlighting a typing error in the title of figure 2 which now reads “Graphical display of QUADAS-2 results”
RESULTS AND DISCUSSION
I do not think it is necessary to keep Figures 5, 8, and 11.
I also suggest the authors consider removing (or reducing) section 3.5 (found in lines 296-363) and Table 4.
Response: We thank the reviewer for their valuable feedback. We agree that the sROC curves in figure 5, 8 and 11 and superfluous and have been removed. We have carefully considered your suggestion to remove section 3.5 which describes other studies faecal biomarkers in Crohn’s disease. We feel that this is a critical section of the manuscript and is something that makes this systematic review and meta-analysis unique as it covers multiple biomarkers in the literature within one article which aids readers who wish to cross compare faecal biomarkers. It is thus our preference to retain this section due to the perceived value to readers. The title of the manuscript has also been updated to reflect the inclusion of this section.
CONCLUSION
This section is adequately described.
Response: We thank the reviewer for their valuable feedback on the conclusion section.
REFERENCES
As pointed out above, I suggest including newer references in the Introduction section and along with the text.
Response: We thank the reviewer for their valuable feedback and have noted the improved the citations to inclusion of more recent publications for the requested sections.
We thank you for raising these issues with us and providing the authors to opportunity to reflect on our analysis and further improve the quality of our manuscript.
Regards
Dr. Anuj Bohra